# Study on distribution characteristic of tourism attractions in international cultural tourism demonstration region in South Anhui in China

**Jing Xu**, **Pengfei Wang***

School of Architecture and Engineering, Huangshan University, Huangshan, China

* pfwang@hsu.edu.cn

## Abstract

Taking data of tourism attractions in international cultural tourism demonstration region in South Anhui in China, this study summarized the distribution characteristics of tourism attractions in region by applying GIS spatial analysis method such as nearest neighbor distance index and kernel density estimation method, and explored how natural conditions, urban infrastructure, social and economic development affect the distribution in order to better understand the distribution of regional tourism resources and serve the direction of tourism development. The study found that the tourist attractions in the demonstration area have a significant agglomeration on the whole, with Huangshan City as the main center and other districts and counties as the sub centers, presenting the trend of hierarchical development. In different kinds of tourism attractions, the distribution of natural tourism attractions and rural pastoral tourism attractions highly related to the topography. And the distribution of modern recreational tourism attractions and humanistic tourism attractions is closely related to hydrology. In terms of urban infrastructure, modern recreational tourism attractions hold the best accessibility with transportation. The core area of the demonstration region include Huangshan City, Chizhou city and Xuancheng city has better tourist industry development. Tourist attractions in other areas are relatively scarce, but the hold great potential for development in the future.

## Introduction

Tourist attractions are the core productivity elements of the tourism industry and the key to the regional tourist attraction [1]. There is a clear causal relationship with the development of regional tourism. As an important part of the tourism system, tourist attractions bear the material carrier of tourism supply and are the basic material conditions for the development of regional tourism. The level and density of tourist attractions are closely related to the regional tourism [2]. Its spatial structure not only affects the spatial behavior of tourists and the realization of economic benefits [3–5], but also has a profound impact on the regional tourism development strategy which is the basis for guiding regional tourism planning, development and management [6–8]. At early stages researches on tourism spatial structure mainly focused on quantitative analysis methods and analytical models [9]. Some classic methods such as central

**Data Availability Statement:** All relevant data are within the paper and its Supporting Information files.

**Funding:** Funded by the Anhui University Humanities and Social Sciences Research Project (SK2018A0389), the Anhui Tourism Talent Training Demonstration Base Project (YYRCZD1703) and the Anhui Quality Engineering Project (2020jyxm1761). The funders had no role in study design, data collection and analysis, decision to publish, or preparation of the manuscript.

**Competing interests:** The authors have declared that no competing interests exist.

place theory, core-periphery model, location theory, and tourism geographic information models and other classic theories have been widely used in the study of spatial structure of tourist attractions. For example, Christaller [10] first applied the theory of central place to the study of the spatial structure of tourist attractions, and found that when tourists are engaged in tourism activities, they move to the periphery of the city and form a certain extent of expansion. Hills and Britton [11, 12] pointed out that the tourist area can be divided into two parts: the core area and the fringe area, and the core area occupies a dominant position in tourism behavior. Milne [13] first applied the geometric analysis method of fractal theory to the study of the spatial distribution characteristics of tourist attractions. Smith [14] specifically summarized relevant theories and empirical methods on the characteristics of tourism spatial structure, including nearest neighbor index, compactness index, and average center point distance. Joanne et al. [15] used Geographic Information System (GIS) technology to analyze the use of national park channels, marking the further use of GIS technology in tourism spatial analysis problems. Kang S et al. [16] studied the spatial structure of the tourist attraction system in Seoul, South Korea based on GIS technology. The results showed that examining the spatial structure of the tourist attraction network is to better formulate competitive tourist destination planning, development and management strategies. Sugimoto K [17] used Global Positioning System (GPS) tracking technology and GIS in the Ueno area of Tokyo, Japan to study the relationship between tourist mobility and urban spatial structure. The results of the research can provide information for local urban management policies and planning. Naranpanawa N et al. [18] examined the spatial linkages between natural amenities and tourism employment spillovers under alternative neighborhood structures. This study used data for Queensland, Australia, however the approach and conclusions can be extrapolated to similar economies.

These classic theoretical methods and models have become common models for subsequent scholars to study the spatial structure of tourism, and have laid a good foundation for in-depth research [19]. To a certain extent, they have promoted the study of the spatial structure of tourist attractions from traditional qualitative description to quantitative analysis. With the widespread use of GIS, the spatial, multi-source and comprehensive characteristics of its data [20] provide a more intuitive and scientific analysis method for the study of tourism spatial structure, which can better assist the use and display of related theoretical models. The use of GIS technology to study the spatial structure of specific types of tourist attractions has a large number of results, and the research objects are rich. There are researches on the spatial distribution of A-level tourist attractions. Pan J et al. [21] took China's A-level tourist attractions as an example to analyze their spatial distribution characteristics across the country. Zhenjie L et al. [22] revealed the evolution of spatial distribution and regional accessibility of A-level tourist attractions in Guangdong Province of China. There are also studies for specific types of tourist attractions, such as historical and cultural tourist attractions, leisure tourist attractions etc. For example, Abdelkader A [23] studied the land use change of Umm Qais, a typical historical tourist attraction in Jordan, and put forward relevant policy suggestions. Gao N et al. [24] used Moran index and panel vector autoregressive model to analyze the spatiotemporal characteristics and influencing factors of China's red tourism network. Jiao J B [25] and Ai Z [26] both used GIS technology to analyze the current situation of traditional Chinese villages and rural tourism, and provide scientific suggestions for the protection, development and utilization of tourism resources in the future. In terms of leisure tourism, Meliha A et al. [27] used GIS method to analyze the tourism terrain, and determined the basic source of tourism and leisure activities in Yahyalı. Bödeker M et al. conducted a city-wide, GIS-based multilevel study on neighborhood walkability and active travel in Germany [28]. Both Czepkiewicz M [29] and Maria T F [30] explored leisure tourism in different cities. After summarizing the relevant research results, it is found that from the perspective of research scale, there are more researches

at the national and provincial scales but less at the regional scale, especially in culturally relevant meso-scale regions. Because of the similar types of this kind of regional resources, the competition among them is more intense. In order to scientifically coordinate the development of such regional tourism industries, it is necessary to conduct detailed research on the spatial distribution characteristics and influencing factors of tourist attractions.

The southern Anhui region of China has superior geographical conditions, excellent ecological environment, profound cultural heritage and rich tourism resources [31]. In order to reflect the advantages of tourism resources in Southern Anhui better, the International Cultural Tourism Demonstration Region in South Anhui (referred to as the Demonstration Region in the following articles) was formally established with the approval of the Chinese government in 2014. It covers seven cities including Huangshan, Chizhou, Anqing, Xuancheng, Tongling, Ma'anshan, and Wuhu, and a total of 47 counties (cities and districts). The region covers an area of 57,000 square kilometers, among which Huangshan, Chizhou, Anqing and Xuancheng City are the core areas of the demonstration region, and it is the first cultural tourism demonstration region set up at the national level in China [32]. The demonstration region is a tourist resource enrichment region which occupies a leading position in the tourism industry of Anhui Province. At the same time, it is a cultural tourism region with important influence and distinctive characteristics in the country and the world [33], so it is an ideal place to study the problem of this article.

After summing up, it is found that the tourism development status of the demonstration region is in good condition overall, but there are still many practical problems such as obvious differences between different areas. Moreover, it is found that there are fewer studies on other tourist attractions except the A-level tourist attractions. At the same time, there are still gaps in the analysis of factors affecting spatial distribution. In this study, POI tourist attractions in the demonstration region were used as data sources, we used GIS and other geospatial analysis tools to analyze the spatial differentiation characteristics of the tourist attractions in the demonstration region, and the reasons for spatial differentiation were discussed in combination with related factors of nature and humanity, which can provide reference for the future tourism development direction and industrial layout of the demonstration region.

## Data source and research methods

### POI data of the tourist attractions in the demonstration region

This research obtain data related to tourist attractions in the demonstration region by using the Gaode map platform and python language technology. The data conform to the definition of tourist attractions and mainly include attributes such as ID, tourist attraction name and latitude and longitude coordinate information. As the essence of tourism resources, A-level tourist attractions are the main source of attraction for local tourism. In order to conduct a more comprehensive analysis and research, other types of tourist attraction data were selected into the article, taking into account the diversity of modern tourism demand and the breadth of tourist attraction. After checking and filtering the data, select several types of data as shown in the table (Table 1), and finally there are 722 tourist attractions.

The selected data was further classified based on the core tourist attractions and the status quo of tourist attractions in the demonstration region by referring to the group standards of the "Tourist Attraction Classification" [34], issued by the China Tourist Attractions Association in December 2019. There are four types of tourist attractions including natural tourism attractions (87), humanistic tourism attractions (121), rural pastoral tourism attractions (420) and modern recreational tourism attractions (94). In this article, WGS1984 spatial coordinates were adopted, and the classified data were spatialized and visualized.

Table 1. Data screening and classification of tourist attractions in demonstration region.

| Type of the tourist attractions | Numbers | Explanation |
|---|---|---|
| A-level tourist attractions | 228 | Including nationally certified 5A, 4A, 3A, 2A tourist attractions |
| Important heritage site under state protection | 62 | Including ancient tombs, ruins, ancient buildings et al |
| Museum, memorial et al. | 77 | 13 of them are duplicated with A-level tourist attractions |
| Historic towns, famous villages and traditional villages | 379 | 11 of them are duplicated with A-level tourist attractions |

Data from screening and sorting of authors.

## Data of other spatial elements

Other spatial element data mainly include natural and humanistic data. (Table 2) showed that the data sources, among them the resolution of DEM elevation data is 30*30 m, the main water system was selected for hydrology, and the two main transportation facilities types of transportation are high-speed and provincial roads. What needs special explanation is that considering the impact of the COVID-19 on the tourism industry, the GDP data was selected from the 2019 China County Statistical Yearbook. For the consistency of the statistical results, the administrative divisions also selected the 2019 data accordingly.

## Research methods

From a spatial perspective, the distribution pattern of any geographic element presents three types: uniform type, random type and cluster type [35, 36] Generally, the approximate distribution of point geographic elements can be directly observed from the map, but it needs to be analyzed in depth and the type of distribution needs to be carried out with the help of other geostatistical methods. This study selected the "Nearest Neighbor Indicator" (NNI), a spatial quantification method that calculates the ratio of the actual nearest neighbor distance (the average value of the distance between each spatial element and its nearest neighbor) to the theoretical nearest neighbor distance [37, 38] to determine the spatial data distribution characteristics of tourist attractions.

$$\overline{d_E} = \frac{1}{2\sqrt{\frac{n}{A}}}, \overline{d_O} = \frac{\sum_{i=1}^{n} d_i}{n}, NNI = \frac{\overline{d_O}}{\overline{d_E}}$$

Among them, $d_E$ is the expected average distance, $d_o$ is the average observation distance, $n$ is the number of tourist attractions, $A$ is the area of the study area, $d_i$ is the distance between

Table 2. Data and data sources.

| data type | data name | data format | data source |
|---|---|---|---|
| Natural factor data | DEM elevation | raster | Geospatial Data Cloud |
|  | hydrology | vector | Edited from Open Street Map data |
| Humanity factor data | administrative divisions | vector | Edited from Open Street Map data |
|  | transportation facilities | vector | Edited from Open Street Map data |
|  | GDP | text | Regional Statistical Yearbook |

Data from sorting of authors.

the element and its closest neighbor, and *NNI* is the closest index. When *NNI* = 1, it means that the tourist attractions are randomly distributed. When *NNI* < 1, it means that the tourist attractions are in agglomeration distribution. The smaller the *NNI* value, the stronger the agglomeration is. When *NNI* > 1, it means that the tourist attractions are evenly distributed.

On this basis, with the help of Kernel Density Estimation (KDE) method [39] was used to calculate the density of tourist attractions in each spatial unit for each spatial element within the study area, it is also possible to analyze the changes in the spatial density of the data further [40]. The basic principle is centered on the location of each spot tourist attraction, and calculate the density of each element within the research range (a circle with radius *r*) of each spatial element through the formula. The formula is as follows.

$$f(x, y) = \frac{1}{nr^2} \sum_{i=1}^{n} k\left(\frac{d_i}{n}\right)$$

Among them, $f(x, y)$ is the density estimation at position $(x, y)$, $n$ is the number of observations, $r$ is the bandwidth, $k$ is the kernel function, $d_i$ is the distance between the position and the $i$-th observation position.

In the process of analyzing the impact of tourist attractions and influencing factors in the demonstration, buffer analysis [41] was used to automatically build a certain range of surrounding polygons around relevant points, lines, and areas, which are used to characterize the influence range of the entity on the neighborhood [42]. According to the format and characteristics of different data, the corresponding vector or raster calculation method was selected to jointly complete this research.

## Spatial differentiation characteristics of tourist attractions in the demonstration region

### The overall distribution characteristics of tourist attractions

The acquired POI tourist attraction data were visualized through projection conversion on ArcGIS 10.6 software platform (Fig 1).

From the perspective of city area, tourist attractions are concentrated in Huangshan City and Xuancheng City, especially Huangshan City. The number of tourist attractions has reached 379, accounting for 52.49% of the total. Huangshan City also has the largest proportion of different types of tourist attractions, especially rural pastoral (279) and modern recreational tourism attractions (44), accounting for 66.43% and 46.81% of the total respectively. After in-depth analysis, it is found that a large number of village-type landscapes in rural pastoral tourism attractions, especially traditional villages, are concentrated in Huangshan City, and most of the museum-type tourist attractions in modern recreational tourism attractions are also gathered here. There are also a large number of traditional villages in the area south of Xuancheng City adjacent to Huangshan City, so the proportion of rural pastoral tourism attractions (80) is also very high.

### Spatial structure characteristics of tourist attractions

Based on the research purpose, the spatial structure characteristics of different types of tourist attractions were discussed. The nearest neighbor index was calculated by using the average nearest distance and theoretical nearest distance of different types of tourist attractions with the help of GIS. It can be seen from the statistical results (Table 3) that the tourist attractions in the demonstration region all present agglomeration distribution in space, but the degree of agglomeration is different. Among them, modern recreational tourism attractions have the

| District | Number |
|---|---|
| Huangshan City | 379 |
| Xuancheng City | 136 |
| Chizhou City | 67 |
| Wuhu City | 55 |
| Tongling City | 28 |
| Anqing City | 19 |
| Maanshan City | 38 |

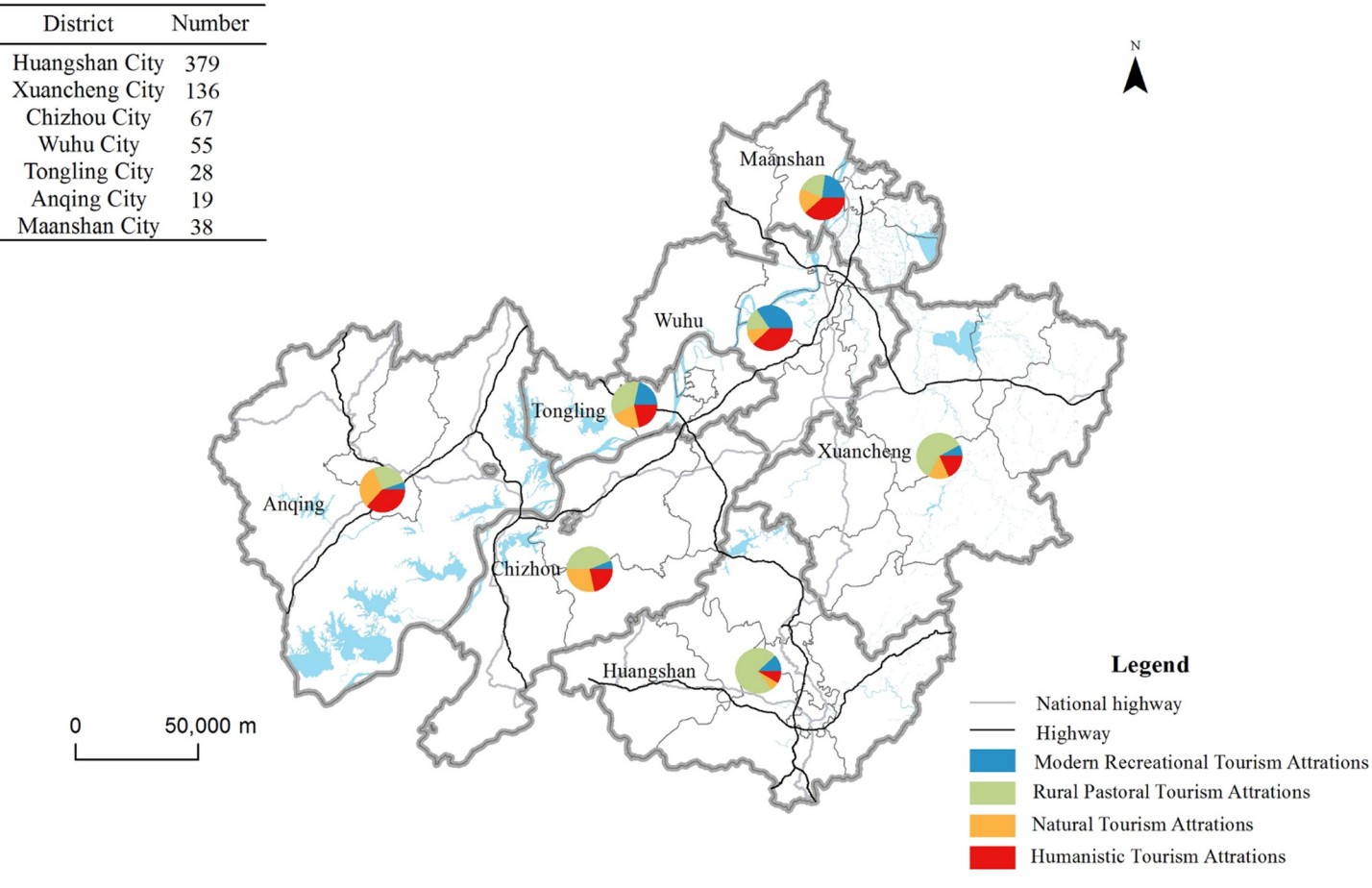

**Fig 1. Regional distribution characteristics of tourist attractions in the demonstration region.** Contains information from OpenStreetMap and OpenStreetMap Foundation, which is made available under the Open Database License.

strongest agglomeration, followed by rural pastoral and humanistic tourism attractions, and natural tourism attractions have the lowest agglomeration.

## Distribution characteristics of hot spots in tourist attractions

The kernel density index can be used to calculate the density of the research elements in the surrounding area, so as to reveal the hot spot distribution characteristics of the elements. The

**Table 3. The agglomeration discrete characteristics of tourist attractions in the demonstration region.**

| Type | Discrete characteristics of spatial agglomeration | | | |
|---|---|---|---|---|
| | Index | Z value | P value | Space state |
| All tourist attractions | 0.508494 | -25.265525 | <0.01 | Agglomeration distribution |
| Modern recreational tourism attractions | 0.460581 | -10.005091 | <0.01 | Agglomeration distribution |
| Rural pastoral tourism attractions | 0.573107 | -16.736870 | <0.01 | Agglomeration distribution |
| Natural tourism attractions | 0.840231 | -2.850902 | <0.01 | Agglomeration distribution |
| Humanistic tourism attractions | 0.591444 | -8.597551 | <0.01 | Agglomeration distribution |

Data from of spatial geostatistics calculation of GIS platform.

results of nuclear density analysis of tourist attractions in the demonstration region (Fig 2) showed that there are differences in the spatial distribution of different types of tourist attractions in different regions. The overall distribution situation is "multi-level centers (each city), hierarchical development". Among them, the area with the highest density of tourist attractions is Huangshan City, especially She and Yi County and Tunxi District are the concentrated areas of tourist attractions, followed by Xuancheng and Chizhou City, and the rest of the area has a lower density. From the perspective of different types of resources (Fig 3), the distribution of natural tourism attractions is relatively scattered. They are mainly distributed in the Huangshan Mountain in the north of Huangshan City and the Jiuhua Mountain in the southeast of Chizhou City. It shows a trend of decreasing towards the periphery with the junction of Qianshan, Wuhu and Tongling City as the multi-core. Rural pastoral tourism attractions are concentrated in Huangshan City, especially in Yi and She County, and the agglomeration effect is obvious. The distribution of modern recreational and humanistic tourism attractions are similar, and they are concentrated in the central city and surrounding areas.

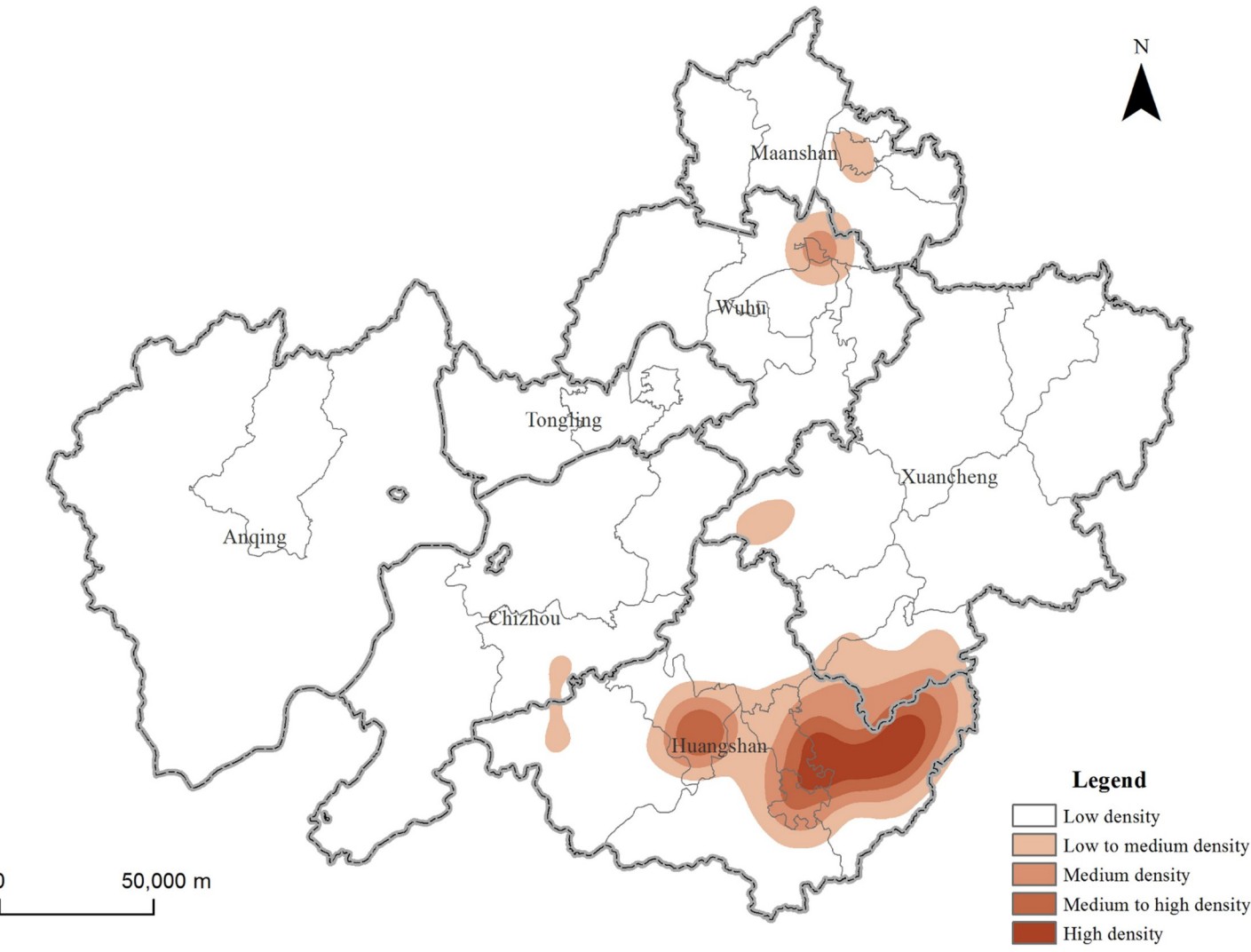

**Fig 2. The overall hot spot distribution characteristics of tourist attractions in the demonstration region.** Contains information from OpenStreetMap and OpenStreetMap Foundation, which is made available under the Open Database License.

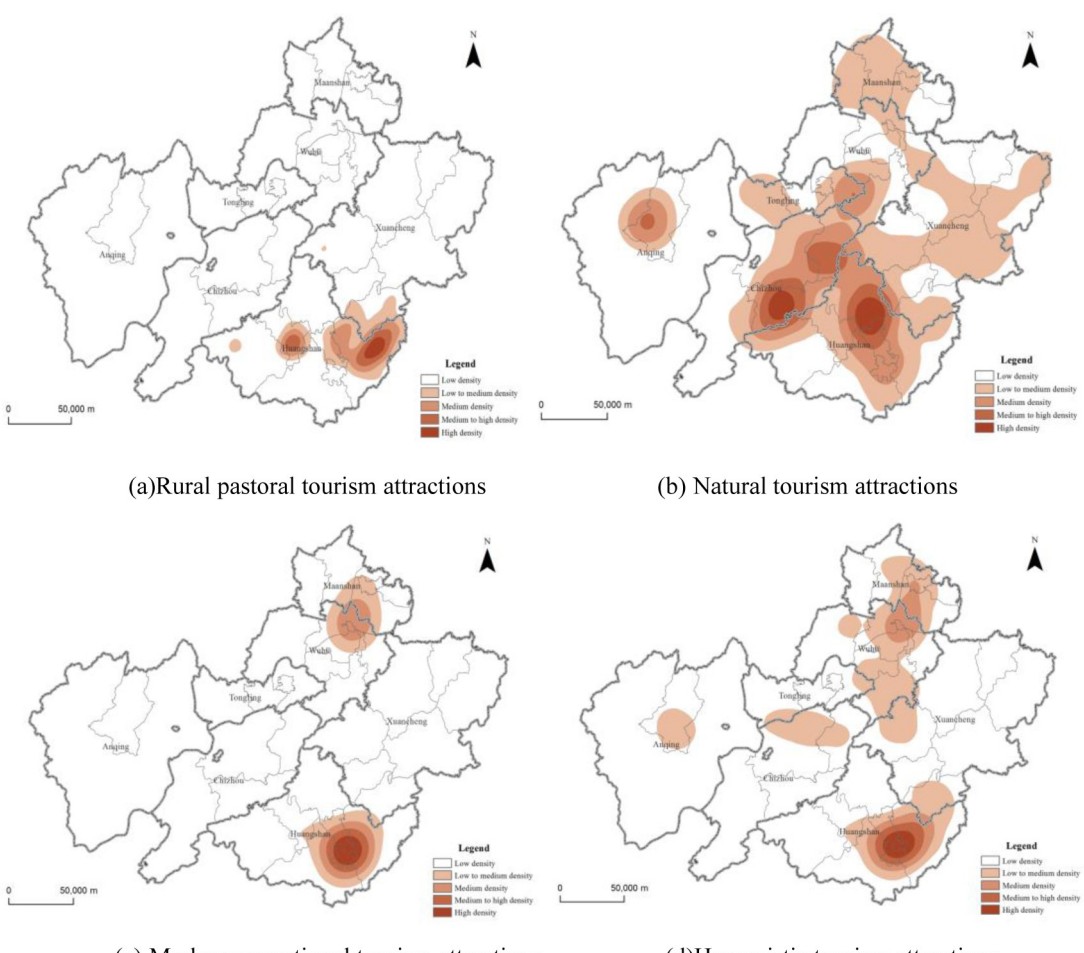

**Fig 3. The hot spot distribution characteristics of tourist attractions in the demonstration region.** Contains information from OpenStreetMap and OpenStreetMap Foundation, which is made available under the Open Database License. (a) Rural pastoral tourism attractions. (b) Natural tourism attractions. (c) Modern recreational tourism attractions. (d) Humanistic tourism attractions.

## Tourist attractions of demonstration region and impact analysis of influencing factors

### Tourist attractions and natural factors

Natural geographic factors affect the formation of tourist attractions, and at the same time have a key impact on the natural characteristics of the tourist attractions [43]. This study selected two basic physical geographic factors topography and hydrology to analyze the impact of tourist attractions. The main terrain of the demonstration region is mainly mountains and hills. The DEM data of the study area was extracted from the elevation information, and the tourist attractions with different elevations were analyzed by mathematical statistics (Fig 4). Modern recreational and humanistic tourism attractions are mainly distributed in plains and hilly areas that no more than 200 meters. Only a few ancient road-type tourist attractions among the humanistic tourism attractions are higher in altitude, and the average altitude of the others is generally less than 100 meters. Rural pastoral tourism attractions are mainly distributed at an altitude of 200–300 meters, and only a few traditional villages have an altitude of more than 500 meters. The average altitude of natural tourism attractions is the highest,

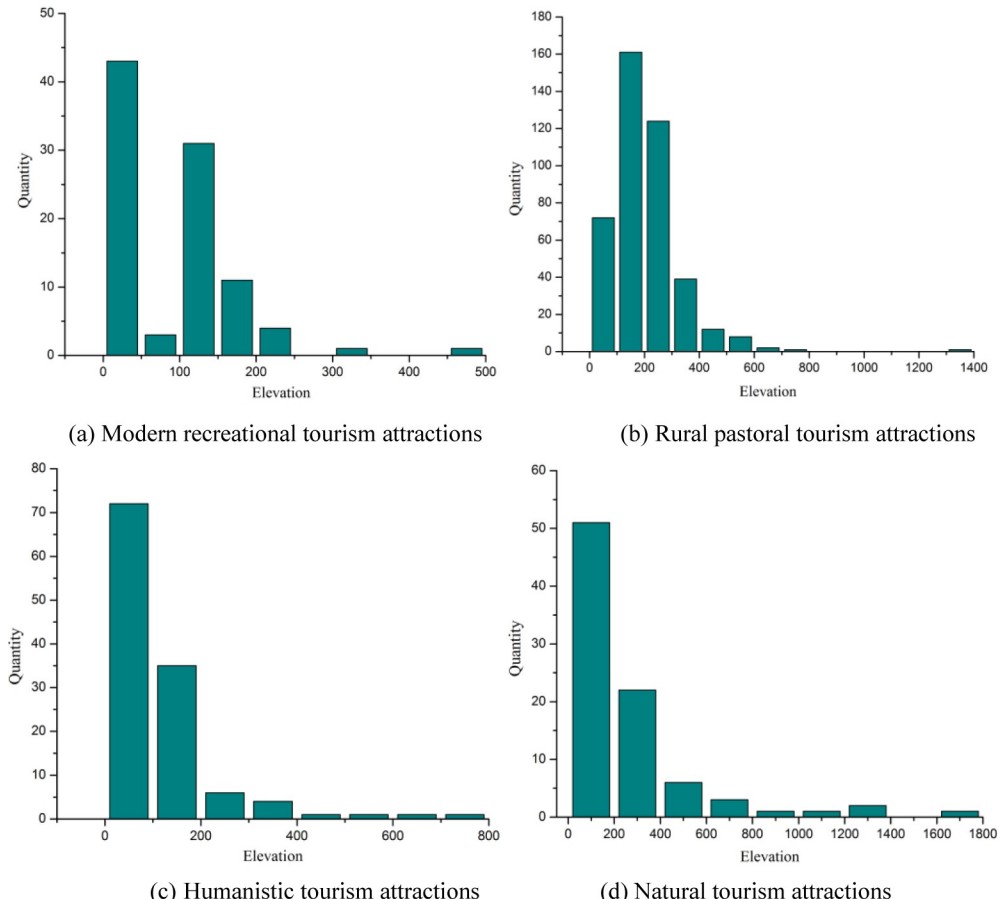

**Fig 4. The elevation frequency distribution of tourist attractions in the demonstration region.** (a) Modern recreational tourism attractions. (b) Rural pastoral tourism attractions. (c) Humanistic tourism attractions. (d) Natural tourism attractions.

reaching 245.94 meter, which also reflects the typical mountain tourism characteristics of the study area from another perspective [31]. In the demonstration region, the mountain tourist attractions represented by Huangshan Mountain, Jiuhua Mountain and Tianzhu Mountain. They are well-known and play an important role of demonstration and guidance in their respective regions.

From a hydrological point of view, hydrological conditions will affect topographic features and further affect the distribution of tourist attractions. Several cities in the demonstration region are located in the Wanjiang urban belt with abundant water resources. In the 2km buffer zone of the main water system, the number of tourist attractions reach 273, accounting for 37.81% of the total. Among them, modern recreational tourism attractions account for the largest proportion, followed by humanistic tourism attractions, rural pastoral tourism attractions and natural tourism attractions. In-depth analysis, the Xin'an River Basin has the strongest concentration of tourist attractions. Xin'an River is the main water system of ancient Huizhou, and is the gathering place for life and production on both sides of the strait which has a large number of historical and cultural relics. Secondly, there are also a large number of tourist attractions in the upper reaches of the Wanjiang River, Qingyi River, Shuiyang River and Qianhe River, reflecting the highly coupled relationship between the water system and tourist attractions (Fig 5).

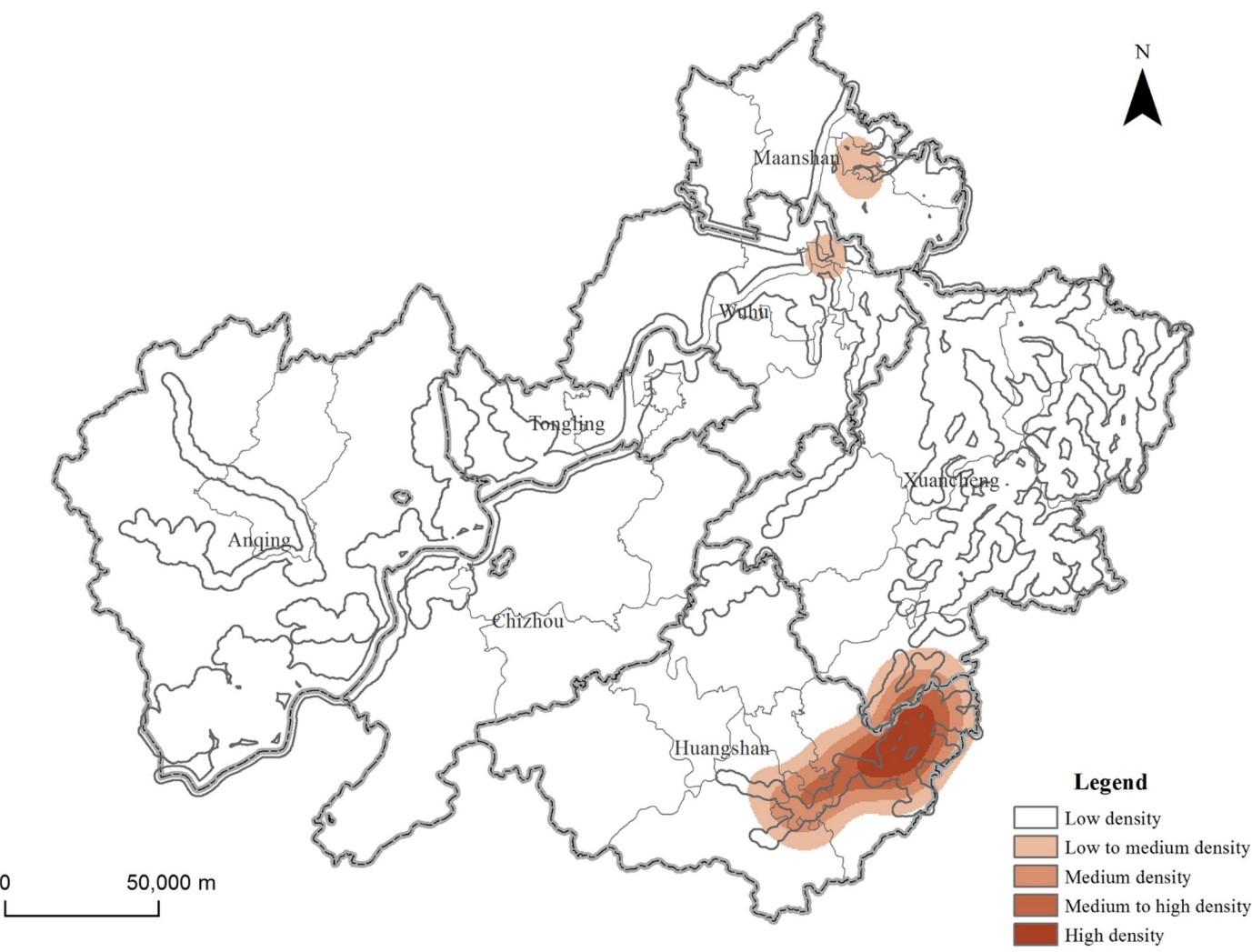

**Fig 5. Analysis of tourist attractions and hydrological factors in the demonstration region.** Contains information from OpenStreetMap and OpenStreetMap Foundation, which is made available under the Open Database License.

## Tourist attractions and infrastructures

The administrative center of a region is the political, economic and cultural center of the region, with convenient transportation and complete infrastructure. It can be seen from the hotspot analysis map of all tourist attractions (Fig 3) that tourist attractions are often formed around the administrative center. Therefore, after spatialization of each administrative center in the demonstration region, a buffer analysis within 5km range was carried out to calculate the proportion of different types of tourist attractions within the buffer zone.

As shown in (Table 4), there are 25.90% of tourist attractions in the statistical unit, of which 86.17% are modern recreational tourism attractions, indicating that this type of tourist attraction is more dependent on the convenient municipal facilities of the administrative center, with developed transportation and location. Humanistic tourism attractions account for 52.89%. Many ancient architecture and historical relic tourist attractions are located in the administrative center of the original ancient Huizhou, that is, the ancient Huizhou counties during the Ming and Qing Dynasties, such as She, Yi and Xiuning County and Jixi District.

**Table 4. Proportion composition of different types of tourist attractions in the demonstration region.**

| Type of tourist attractions | Number | Proportion |
|---|---|---|
| Modern recreational tourism attractions | 81 | 86.17% |
| Natural tourism attractions | 7 | 8.05% |
| Rural pastoral tourism attractions | 35 | 8.33% |
| Humanistic tourism attractions | 64 | 52.89% |
| All tourism attractions | 187 | 25.90% |

Data from buffer analysis statistical calculation of GIS platform.

Rural pastoral and natural tourism attractions are relatively less affected by the administrative center. The former contains a large number of traditional villages, while the traditional villages in Huizhou are concentrated in the range of 0–35 km from the central town [44], and the latter is more dependent on natural resources, so the proportion is the smallest.

The research further selected the most basic transportation facilities for analysis. Transportation is an important bridge connecting tourist source and destination [45, 46]. The overall transportation in the demonstration region is convenient. The main traffic arteries (high-speed, national highway) in the study area were selected to establish a 5km buffer zone. Statistics (Fig 6) showed that there are 331 tourist attractions distributed in the buffer zone, accounting for 45.34% of the total number of tourist attractions. Among them, modern recreational tourism attractions (73) account for the highest proportion, reaching 73%, which is consistent with the above analysis results. Followed by humanistic and rural pastoral tourism attractions, accounting for 58.54% and 37.39% respectively. The largest number of tourist attractions are clustered on the G205 National Highway, with the number reaching 119, accounting for 16.30% of the total number of tourist attractions. There are also a large number of tourism attractions clustering around the G3, G5011, and 318 national highways, which shows that traffic has a profound impact on the distribution of tourist attractions.

## Tourist attractions and economic development

GDP is an important indicator to measure regional economic development, and it also affects the distribution of regional tourist attractions to a certain extent [47]. This study attempted to analyze the proportion of tourism industry in regional economic development by studying the relationship between the two. Reclassify and assign values based on the density data of tourist attractions in the demonstration region and the distribution of GDP in districts and counties according to 1–5 points, then do raster subtraction. The greater the absolute value is, the greater the relative difference between the two. A positive value indicates that the region's tourism development has a higher status in the region, while a negative value is the opposite. The analysis found that there is obviously a superior circle of tourism development in the demonstration region in which Huangshan City is located in the superior area of tourism development. The regional tourism and economic development of most parts of Huangshan City, bordering Chizhou and Xuancheng City are relatively balanced. In particular, Yi and She County of Huangshan City have the most prominent advantages in tourism development. The central and northern parts of Xuancheng City, the northwestern part of Chizhou, Tongling, Wuhu and Ma'anshan City all have areas with weak tourism development. In particular, Wuwei County and parts of Jiujiang District of Wuhu City, parts of Xuanzhou District of Xuancheng City, parts of Dangtu County of Ma'anshan City and other areas are lagging behind in tourism economy. There are traditional industries in these areas which have fewer tourist attractions and the development of the tourism industry is relatively backward. It can

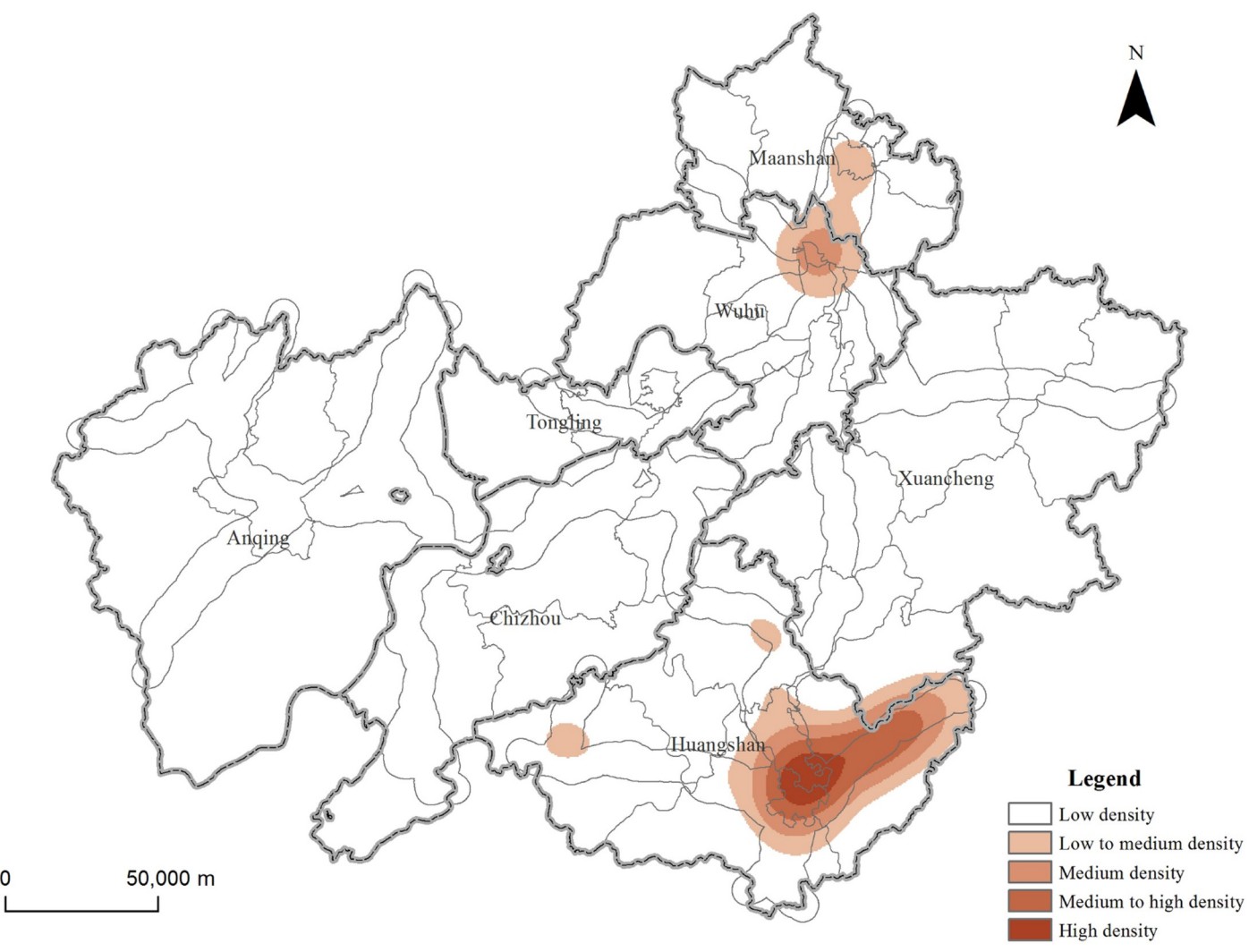

**Fig 6. Analysis of tourist attractions and traffic factors in the demonstration region.** Contains information from OpenStreetMap and OpenStreetMap Foundation, which is made available under the Open Database License.

be concluded that the core areas in the demonstration region, including Huangshan, Chizhou, and Xuancheng City have formed cultural tourism as a leading industry. Other cities regard the cultural tourism industry as an important pillar industry for regional economic development with a large industrial contribution rate (Fig 7). The cultural tourism industry has become an important engine and booster for regional economic development.

## Conclusions

The advantage of using spatial geostatistics to analyze the spatial characteristics of tourist attractions is that certain abstract features can be quantified [48, 49]. After evaluating and analyzing the spatial characteristics of different types of tourist attractions in the International Cultural Tourism Demonstration Region in South Anhui, the following conclusions are drawn: Huangshan City is the main center of the tourist attractions in the demonstration region, and Chizhou and Xuancheng City are the secondary centers, presenting the trend of

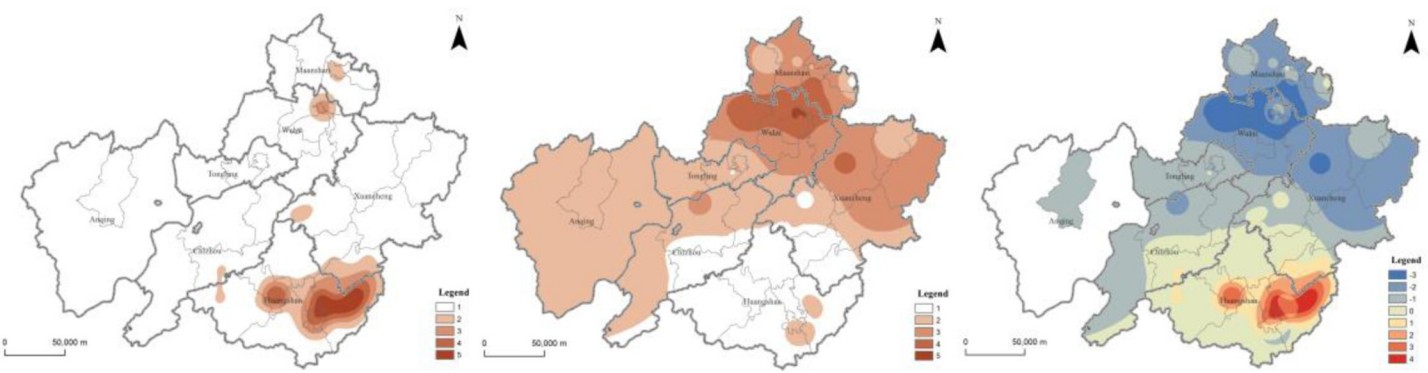

(a)GPD classification    (b)Tourism attractions density classification    (c)Raster operation result classification

**Fig 7. Analysis of tourist attractions and GDP factors in the demonstration region.** Contains information from OpenStreetMap and OpenStreetMap Foundation, which is made available under the Open Database License. (a) GPD classification. (b) Tourism attractions density classification. (c) Raster operation result classification.

hierarchical development. The distribution characteristics of different types of tourist attractions all present agglomeration distribution but the degree of agglomeration is different.

1. The natural environment is an important environmental foundation for the tourist attractions in the demonstration region. Among them, elevation is particularly important for natural and rural pastoral tourism attractions. It will affect the distribution of typical geography and traditional village tourist attractions, while modern recreational tourism attractions are significantly affected by water systems.

2. Infrastructure is the key support of the tourist attractions. Different types of tourist attractions have different dependence on infrastructure. Modern recreational and humanistic tourism attractions are highly dependent, and tourist attractions are distributed in a large number of important traffic arterial buffer zones. Among them, modern recreational tourism attractions have the highest dependence on traffic.

3. The development of tourism is uneven in the demonstration region. The core areas of the demonstration region, including the cities of Huangshan, Chizhou and Xuancheng have relatively developed tourism industries, while the cities of Wuhu, Ma'anshan, and Tongling which are dominated by traditional industries have relatively few tourism resources and the tourism industry is backward. We should actively explore leisure tourism products and regard tourism as an important supplement to economic development.

## Discussion

1. Through analysis, it is found that the tourist attractions in the demonstration region are dense and high-level, but they belong to different administrative divisions, and there is a phenomenon of "self-administration", which affects the tourism development of the demonstration region. In terms of management, a regional coordination committee should be established to strengthen overall coordination, break the barriers of administrative space, and achieve integrated development. At the same time, clarify the tourism characteristics of various regions and formulate tourism development policies in accordance with local conditions. For example, Huangshan, Chizhou, and Xuancheng City should dig deep into local culture, create unique and charming cultural tourism products, and lead the urban tourism linkage and development in the periphery of the demonstration region. Wuhu, Tongling,

and Ma'anshan City can build on the advantages of urban economic development and provide strong backing for tourism development with capital and technology. In short, through the optimization and coupling of space, design high-quality tourism products, enrich the tourism industry, and promote the development of tourism in the demonstration region. In general, we should strive to optimize and integrate space, design high-quality tourism products, enrich tourism formats, and further promote the development of tourism in the demonstration region.

2. This study selected representative POI tourist attractions data, while ignored no-level tourism resources. In order to reflect the integrity of regional tourism development, in-depth analysis is needed in the implementation of tourism development in the demonstration region to explore potential tourist attractions, to give play to the overall advantages, and to serve the tourism development of the demonstration region.

3. There are many factors that affect the distribution of tourist attractions. The study only selected three categories and five races of natural factors, infrastructure, and economic development for correlation analysis, which is the limitation of this study. Therefore, the scope of data can be expanded in the future, and its process and influence mechanism can be further discussed. The spatial distribution characteristics bring the resource foundation for the industrial positioning of regional tourism development, and the type distribution characteristics provide the basis for the development direction of local tourism products and industries.

## Supporting information

**S1 Data.**
(RAR)

## Author Contributions

**Formal analysis:** Pengfei Wang.

**Writing – original draft:** Jing Xu.

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
