## [Decision Letter · Decision Letter 0]

20 Dec 2021

PONE-D-21-36157Study on Distribution Characteristic of Tourism Attrations in International Cultural Tourism Demonstration Region in South Anhui in ChinaPLOS ONE

Dear Dr. Xu,

Thank you for submitting your manuscript to PLOS ONE. After careful consideration, we feel that it has merit but does not fully meet PLOS ONE’s publication criteria as it currently stands. Therefore, we invite you to submit a revised version of the manuscript that addresses the points raised during the review process.

We look forward to receiving your revised manuscript.

Kind regards,

Lóránt Dénes Dávid, PhD

Academic Editor

PLOS ONE

Journal Requirements:

"Funded by the Anhui University Humanities and Social Sciences Research Project (SK2018A0389), the Anhui Tourism Talent Training Demonstration Base Project (YYRCZD1703) and the Anhui Quality Engineering Project (2020jyxm1761)."

7. We note that Figures 1,2,3,5,6 snd 7 in your submission contain [map/satellite] images which may be copyrighted. All PLOS content is published under the Creative Commons Attribution License (CC BY 4.0), which means that the manuscript, images, and Supporting Information files will be freely available online, and any third party is permitted to access, download, copy, distribute, and use these materials in any way, even commercially, with proper attribution. For these reasons, we cannot publish previously copyrighted maps or satellite images created using proprietary data, such as Google software (Google Maps, Street View, and Earth). For more information, see our copyright guidelines: http://journals.plos.org/plosone/s/licenses-and-copyright.

a. You may seek permission from the original copyright holder of Figures 1,2,3,5,6 snd 7 to publish the content specifically under the CC BY 4.0 license.  

Reviewers' comments:

Reviewer's Responses to Questions

**Comments to the Author**

1. Is the manuscript technically sound, and do the data support the conclusions?

Reviewer #1: Partly

Reviewer #2: Yes

2. Has the statistical analysis been performed appropriately and rigorously? 

Reviewer #1: Yes

Reviewer #2: Yes

3. Have the authors made all data underlying the findings in their manuscript fully available?

Reviewer #1: Yes

Reviewer #2: Yes

4. Is the manuscript presented in an intelligible fashion and written in standard English?

Reviewer #1: No

Reviewer #2: Yes

5. Review Comments to the Author

Reviewer #1: The paper investigates a really interesting topic with an advanced methodology toolset. The title covers the content, abstract is appropriate and compact. Introduction is more or less acceptable, however 1) I recommend to separate the literature review part into a separate chapter, 2) a stronger focus should be set on the context and reasons, importance of the research and on the research questions. Literature review chapter is completely missing in this paper; I recommend to write a separate literature review chapter where the relevant international sources (preferably from international ranked and indexed journals) are processed and analysed critically and in a comparative way. Methodology is well selected and supports the results. Results are clear and well demonstrated. However, the discussion part is missing. Conclusions part is appropriate and based on the results. The conclusions are very useful for the practice and policy decision makers as well, after publishing of this paper it should be promoted to them.

Reviewer #2: The paper deals with a very current topic which is presented in a very accurate way. The structure of the paper is very clear and the results contaiin novelties for the sciences. My suggestions for the authors are the following: Abstract should be extended with the aims and research methods of the paper. The extention of the literature review with some more up-to-date items is recommended. Some suggestions for the future are also recommended.

6. PLOS authors have the option to publish the peer review history of their article (what does this mean?). If published, this will include your full peer review and any attached files.

Reviewer #1: No

Reviewer #2: No

---

## [Author Response · Author response to Decision Letter 0]

12 Feb 2022

Dear Editor

Thanks for your comments on our paper. We have revised the paper according to your comments. The following is the answers and revisions I have made in response to the reviewers’ questions and suggestions on an item by item basis.

Reviewer 1:

Q1: I recommend to separate the literature review part into a separate chapter.

A1: According to your opinion, the author adjusted the second paragraph of the introduction into a separate chapter. The modified part has been marked in red font.

Q2: A stronger focus should be set on the context and reasons, importance of the research and on the research questions. Literature review chapter is completely missing in this paper; I recommend to write a separate literature review chapter where the relevant international sources (preferably from international ranked and indexed journals) are processed and analysed critically and in a comparative way. 

A2: According to your opinion, the author adjusted the structure of the introduction. The introduction was elaborated in four paragraphs. The first paragraph focuses on the background and reasons for choosing the topic and the significance of the research, the second paragraph is adjusted to a separate literature review chapter. The author read the journals from international rankings and indexes carefully to understand the research history and current situation of related issues. The third paragraph summarizes the existing research results, focusing on the shortcomings of the existing research results. The author hopes to make up for the shortcomings through their own research. The fourth paragraph explains the reasons for the research site selected for this study and proposed research questions. The modified part has been marked in red font. 

Q3: However, the discussion part is missing. Conclusions part is appropriate and based on the results. The conclusions are very useful for the practice and policy decision makers as well, after publishing of this paper it should be promoted to them.

A3: According to your opinion, the author added a discussion part at the end of the article, discussing the three aspects of policy recommendations, research deficiencies, and future research directions. The modified part has been marked in red font. 

Reviewer 2:

Q1: Abstract should be extended with the aims and research methods of the paper. 

A1: According to your opinion, the author expanded the purpose and research methods of the abstract part of the paper. The modified part has been marked in red font. 

Q2: The extension of the literature review with some more up-to-date items is recommended. 

A2: According to your opinion, the author read the journals from the international rankings and indexes carefully, and applied it to the second paragraph of the introduction in the first part, expanding the literature review. The modified part has been marked in red font. 

Q3: Some suggestions for the future are also recommended.

A3: According to your opinion, the author added a discussion section at the end of the article, and added suggestions for the future in the first paragraph of the discussion section. 

We would like to express our gratitude again to you and reviewers for the comments on our paper. I am looking forward to hearing from you.

Yours sincerely,

Pengfei Wang

---

## [Decision Letter · Decision Letter 1]

10 May 2022

PONE-D-21-36157R1Study on Distribution Characteristic of Tourism Attractions in International Cultural Tourism Demonstration Region in South Anhui in ChinaPLOS ONE

Dear Dr. Pengfei Wang,

Thank you for submitting your manuscript to PLOS ONE. After careful consideration, we feel that it has merit but does not fully meet PLOS ONE’s publication criteria as it currently stands. Therefore, we invite you to submit a revised version of the manuscript that addresses the points raised during the review process.

We look forward to receiving your revised manuscript.

Kind regards,

Lóránt Dénes Dávid, PhD

Academic Editor

PLOS ONE

Journal Requirements:

Reviewers' comments:

Reviewer's Responses to Questions

**Comments to the Author**

1. If the authors have adequately addressed your comments raised in a previous round of review and you feel that this manuscript is now acceptable for publication, you may indicate that here to bypass the “Comments to the Author” section, enter your conflict of interest statement in the “Confidential to Editor” section, and submit your "Accept" recommendation.

Reviewer #2: (No Response)

Reviewer #3: All comments have been addressed

Reviewer #4: All comments have been addressed

2. Is the manuscript technically sound, and do the data support the conclusions?

Reviewer #2: Yes

Reviewer #3: Yes

Reviewer #4: Yes

3. Has the statistical analysis been performed appropriately and rigorously? 

Reviewer #2: Yes

Reviewer #3: Yes

Reviewer #4: Yes

4. Have the authors made all data underlying the findings in their manuscript fully available?

Reviewer #2: Yes

Reviewer #3: Yes

Reviewer #4: Yes

5. Is the manuscript presented in an intelligible fashion and written in standard English?

Reviewer #2: Yes

Reviewer #3: Yes

Reviewer #4: Yes

6. Review Comments to the Author

Reviewer #2: The article covers a current topic with appropriate methodology and excellent new results. I suggest expanding your literature review.

Reviewer #3: The topic of the paper is actual and very interesting. This paper summarized the distribution characteristics of tourism attractions in one Chinese region. Literatures were cited correctly and were fit to the topic. Used method - GIS spatial analysis method such as nearest neighbour distance index and kernel density estimation method - were correct and helped to analyse the research area. Tables of the paper are informative and help understanding. Paper was improved suggestions by the reviewers.

Reviewer #4: The topic of is very interesting and also relevant. The elaboration and the applied methodology seem accurate. There is still a formal problem: there are no source indications under the tables in the paper. (It has to be done even in case of field research data as well!). The bibliographic review could also be broadened, in a high-ranking journal, like Plos One, it could even be more than 50 sources instead of the present 31. After the mentioned improvements were done, the paper can be recommended to be accepted and published.

7. PLOS authors have the option to publish the peer review history of their article (what does this mean?). If published, this will include your full peer review and any attached files.

Reviewer #2: No

Reviewer #3: No

Reviewer #4: No

---

## [Author Response · Author response to Decision Letter 1]

29 May 2022

Dear Editor

Thanks for your comments on our paper. We have revised the paper according to your comments. The following is the answers and revisions I have made in response to the reviewers’ questions and suggestions on an item by item basis.

Reviewer 2:

Q1: The article covers a current topic with appropriate methodology and excellent new results. I suggest expanding your literature review.

A1: According to your opinion, the author has expanded the literature review. The modified part has been marked in red font.

Reviewer 3:

Thank you for your valuable comments, we will revise according to the comments made by the reviewers. 

Reviewer 4:

Q1: The topic of is very interesting and also relevant. The elaboration and the applied methodology seem accurate. There is still a formal problem: there are no source indications under the tables in the paper. (It has to be done even in case of field research data as well!). The bibliographic review could also be broadened, in a high-ranking journal, like Plos One, it could even be more than 50 sources instead of the present 31. After the mentioned improvements were done, the paper can be recommended to be accepted and published.

A1: According to your opinion, the author has expanded the literature review and broadened the bibliographic review. For the formal problem posed the author has added source indications to all tables in the paper. The modified part has been marked in red font. 

We would like to express our gratitude again to you and reviewers for the comments on our paper. I am looking forward to hearing from you.

Yours sincerely,

Pengfei Wang

---

## [Editor Report · Decision Letter 2]

1 Jun 2022

Study on Distribution Characteristic of Tourism Attractions in International Cultural Tourism Demonstration Region in South Anhui in China

PONE-D-21-36157R2

Dear Dr.pengfei Wang, (may be Pengfei, with capital first letter)

We’re pleased to inform you that your manuscript has been judged scientifically suitable for publication and will be formally accepted for publication once it meets all outstanding technical requirements.

Kind regards,

Lóránt Dénes Dávid, PhD

Academic Editor

PLOS ONE

Additional Editor Comments (optional):

Accepted based on the corrections after the reviewers' opinion.
---

## [Editor Report · Acceptance letter]

9 Jun 2022

PONE-D-21-36157R2 

Study on Distribution Characteristic of Tourism Attractions in International Cultural Tourism Demonstration Region in South Anhui in China 

Dear Dr. Wang:

I'm pleased to inform you that your manuscript has been deemed suitable for publication in PLOS ONE. Congratulations! Your manuscript is now with our production department. 

Kind regards, 

on behalf of

Dr. Lóránt Dénes Dávid 

Academic Editor

PLOS ONE